



# Seasonal and spatial patterns of primary production in a high-latitude fjord affected by Greenland Ice Sheet run-off

Holding, Johnna M.[1*], Markager, Stiig[2], Juul-Pedersen, Thomas[3], Paulsen, Maria L.[1], Møller, Eva F. [3],
Meire, Lorenz[3,4], and Sejr, Mikael K.[1]

[1] Arctic Research Centre, Bioscience, Aarhus University, Ny Munkegade 114, bldg. 1540, DK-8000 Aarhus C, Denmark.
[2] Department of Bioscience, Aarhus University, Frederiksborgvej 399, 4000 Roskilde, Denmark.
[3] Greenland Climate Research Centre, Greenland Institute of Natural Resources, Kivioq 2, 3900 Nuuk, Greenland
[4] Department of Estuarine and Delta Systems, NIOZ Royal Netherlands Institute of Sea Research and Utrecht University,
Yerseke, The Netherlands

*Correspondence to: Johnna M. Holding (johnna@bios.au.dk)





**Abstract**

Primary production on the coast and in Greenland fjords sustains important local and sustenance fisheries. However, unprecedented melting of the Greenland Ice Sheet (GrIS) is impacting the coastal ocean, and its effects on fjord ecology remain understudied. It has been suggested that as glaciers retreat, primary production regimes may be altered rendering fjords less productive. Here we investigate patterns of primary productivity in a Northeast Greenland fjord (Young Sound, 74 N), which receives run-off from the GrIS via land-terminating glaciers. We measured size fractioned primary production during

the ice- free season along a spatial gradient of meltwater influence. We found that, apart from a brief under-ice bloom during summer, primary production remains low (between 50-200 mg C m$^{-2}$ day$^{-1}$) however steady throughout the ice-free season, even into the fall. Low productivity is due to freshwater run-off from land-terminating glaciers causing low light availability and strong vertical stratification limiting nutrient availability. The former is caused by turbid river inputs in the summer restricting phytoplankton biomass to the surface and away from the nitracline. In the outer fjord where turbidity plays less of

a role in light limitation, phytoplankton biomass moves higher in the water column in the fall due to the short day- length as the sun angle decreases. Despite this, plankton communities in this study were shown to be well adapted to low light conditions, as evidenced by the low values of saturating irradiance for primary production (5.8 – 67 µmol photons m$^{-2}$ s$^{-1}$). With its low but consistent production across the growing season, Young Sound offers an alternative picture to other more productive fjords which have highly productive spring and late summer blooms and limited fall production. However, patterns

of primary productivity observed in Young Sound are not only due to the influence from land-terminating glaciers but are also consequences of the nutrient deplete coastal boundary currents and the shallow entrance sill, features which should also be considered when generalizing about how primary production will be affected by glacier retreat in the future.



## 1 Introduction

The coastal marine coastal ecosystems around Greenland are currently experiencing rapid changes due to climate warming. The Greenland Ice Sheet (GrIS) is melting at unprecedented rates (Chen et al., 2006; Enderlin et al., 2014), reaching a record melt extent in 2012, where 97% of the total ice sheet area displayed melting (Nghiem et al., 2012). Subglacial discharge as well as ocean warming is causing increased calving rates and the retreat of tidewater glaciers (Howat et al., 2007; Rignot et al., 2010; Straneo and Heimbach, 2013). Though since 2009, 84% of the rapid mass loss of the GrIS is said to be

due to increased surface run-off (Enderlin et al., 2014). In Southern Greenland freshwater discharge into the surrounding coastal region has increased by almost 50% over just the last two decades (Bamber et al., 2012). This reported freshening of fjords and coastal waters around Greenland (Böning et al., 2016; Sejr et al., 2017) has major consequences for the marine ecosystem as well as for the inshore fisheries (Meire et al., 2017). However, the magnitude and direction of these effects on the different fjord ecosystems around Greenland are still largely unclear.


         Studies from tidewater glacial fjords suggest that melting in late summer is beneficial to pelagic primary production as subglacial discharge causes upwelling at the glacier front (Hopwood et al., 2018; Mortensen et al., 2013). This subsidizes plankton communities in the surface layer with fresh nutrients while stratifying the water column (Juul-Pedersen et al., 2015; Krawczyk et al., 2015; Meire et al., 2015, 2017). For example, the tidewater glacial fjord, Godthåbsfjord, on the Southwest

coast of Greenland features both a highly productive spring bloom (1,743 mg C $m^{-2}$ $d^{-1}$; Juul-Pedersen et al. 2015) as well as a second, prolonged, almost equally as productive late summer bloom (1,383 mgC $m^{-2}$ $d^{-1}$; Juul-Pedersen et al. 2015) due to nutrients subsidies derived from upwelling at the glacial front (Meire et al., 2017). Alternatively, other glacial fjords without these upwelling mechanisms report summer melting to introduce nutrient poor freshwater, which dilutes available nutrients in the upper stratified layers (Reisdorph and Mathis, 2015).


         Young Sound is a Northeast Greenland fjord devoid of any glacial upwelling mechanisms. It is a seasonally ice-covered fjord that is influenced by meltwater from the GrIS via land terminating glaciers (Citterio et al., 2017). Thus at the onset of GrIS melt in the summer a shallow freshwater lens is established throughout the fjord (Bendtsen et al., 2007; Rysgaard et al., 1999). In contrast, to Godthåbsfjord, Young Sound records a more moderate spring bloom that is less than a quarter as

productive (< 300 mg C $m^{-2}$ $d^{-1}$) and has low annual pelagic primary productivity (10.3 g C $m^{-2}$ $yr^{-1}$) attributed to the short open-water period (Rysgaard et al., 1999).

         Previously, it was considered that that primary production in Young Sound and other Arctic fjords is proportional to the length of the open water light period (Rysgaard et al., 1999), and that future annual primary production across the Arctic

will increase as the ice-free season lengthens. However recent research on the effects of freshening in the Arctic suggest that an increase in freshwater inputs intensifies stratification and impedes vertical nutrient supply counteracting the effects of a lengthening of the open water season (Bergeron and Tremblay, 2014; Coupel et al., 2012, 2015; McLaughlin and Carmack, 2010; Yun et al., 2016). Furthermore, freshwater input in the coastal Arctic also brings large sediment loads and/or glacier flour, clouding the water column and affecting the primary productivity via light limitation (Wiktor et al., 1998). Murray et

al. (2015) demonstrated a strong relationship of water column turbidity and light attenuation in two Greenland fjords (Godthåbsfjord and Young Sound) with potential implications for primary production. Thus, it is likely that the light





environment for primary producers in Young Sound is affected by both sea ice cover in the spring and run-off in the summer, and the vertical nutrient supply in the fjord is limited by stratification due to freshwater input that lacks a glacial upwelling mechanism.


As Young Sound is one of the locations of the Greenland Ecosystem Monitoring programs (see Christensen et al. (2017) and articles therein), rates of marine primary productivity there have been reported in several articles over the last 20 years (e.g. Rysgaard et al. 1999; Nielsen et al. 2007). This study, however focuses on determining spatio-temporal patterns during one open-water season with the aim of investigating the impact of freshwater input on pelagic primary production

there. Based on previous findings we sought to determine if the low productivity in Young Sound was caused by turbid freshwater input or the short open water period as previously hypothesized. We suggest that future productivity in Young Sound will be constrained by increasing run-off, which both reduces photic zone depth and increases stratification, rather than reduced ice-cover. Furthermore, we discuss how strong stratification and the unique circulation of the fjord limits the renewal of nutrients to the surface water, making this fjord extremely nutrient deplete throughout the productive season. Little is

known about how freshwater run-off in non-tidewater glacial fjords will affect primary productivity, even though 70-50% of freshwater run-off from the Greenland Ice Sheet comes from land terminating glaciers (Enderlin et al., 2014), making this study important as glaciers continue to recede.

## 2 Methods


### 2.1 Study area

The study was conducted in Young Sound, a High-Arctic fjord in Northeast Greenland (74.2 -74.3 °N, 19.7-21.9 °W; Fig. 1). Young Sound is 90 km long, 2 to7 km wide and covers an area of 390 km². The maximum depth of the fjord is 330 m with two shallow sills, the outermost reaches ~45 meters depth and separates the deeper parts of the fjord from the

Greenland Sea. Sampling was conducted at four stations located along a length section from the inner Tyroler fjord arm (Station 1) to the shelf waters on the outer side of the sill (Station 4) (Fig. 1). Glaciers cover ca. 33 % of the drainage area of the fjord and land-terminating glaciers contribute 50-80% of the annual freshwater runoff with highest contributions coming from the Tyroler, Lerbugt, and Zackenberg rivers in the inner fjord (Bendtsen et al., 2014; Citterio et al., 2017).

The four stations were selected to represent a gradient of changing physical conditions in Young Sound. Station 1 in the inner Tyroler fjord, represents the inner fjord section and is impacted by runoff from the Tyroler River. Station 2 is located at the mouth of the Zackenberg River and represents the central part of the fjord also affected by run-off. Station 3 lies midway out of the fjord and is also the standard sampling station for the on-going time series and the location of previous reports of primary productivity in Young Sound (Nielsen et al., 2007; Rysgaard et al., 1999), and finally Station 4 was positioned just

outside of the fjord in the Greenland Sea and reflects shelf conditions (Fig. 1). Throughout the paper we refer to inner fjord stations as Stations 1 and 2 as "inner fjord" and Stations 3 and 4 as "outer fjord". The stations 1, 2, 3 and 4 are also part of the transect monitored by the Greenland Ecosystem Monitoring (GEM) MarineBasis Zackenberg program in which they are named Tyro 05, YS 3.18, Standard Station, and GH 05, respectively.

### 2.2 Sampling



The first sampling was conducted through the sea ice on July 11, 2014 (Julian day 192), through a hole in the ice at Station 3, when only the central part of the fjord was still ice covered (Fig. 1). The ice broke up in the central part on July 15, 2014 and the fjord was rendered ice-free within 24 hours. Subsequently, stations were sampled approximately every 10th day (from July 17–August 10; Julian days 198–222) and then again in the fall period before new ice formation (September 4–October 6: Julian days 247–279). Julian days were used during analyses but replaced by calendar days in figures for simplicity.

After the sea ice break-up, sampling was carried out from the research vessel *Aage V. Jensen*. A Seabird SBE 19+ CTD profiler was deployed at every sampling occasion and recorded vertical profiles of temperature, salinity, chlorophyll *a* fluorescence ($flu_{chl}$; Seapoint), turbidity (Seaspoint; FTU), and photosynthetically active radiation (PAR; $4\pi$ sensor from Biospherical; $\mu mol\ m^{-2}\ s^{-1}$). Water was sampled using a mini rosette with $12 - 1.7$ L Niskin bottles from 6 standard depths (1, 10, 20, 30, 40 and 100 m) and one or two additional depths of florescence maximum (DFM) when this did not overlap with one of the standard depths. The DFM was approximated prior to every sampling using florescence profiles from a Satlantic Free-falling Optical Profiler (Murray et al., 2015). Underwater light was recorded relative to a deck sensor.

Additionally, CTD profiles at approximately 25 stations (Fig. 1a) along the length of the fjord were recorded on four separate occasions during the season : July 25th (Julian day: 206), August 8th (220), September 17th (260) and October 4th (277). For simplicity, only one date from each season (summer and fall—see data analysis section below) was chosen to depict in Fig. 1d-i, as the patterns were visibly similar among dates within the same season.

Light attenuation was estimated from the CTD-profiles using a two-phase Weibull function as described in Murray et al. (2015). This technique insured that the pronounced changes in turbidity with depth was reflected in the light attenuation, which decreased with depth. The photic depth ($Z_p$) was calculated as the last depth from surface with a positive daily primary production, assuming a respiration equal to 5 percent of $P_m$. The mixed layer depth ($Z_m$) was considered as the largest density change below 5m. The stratification index (SI) was determined as the difference between the density at 80 m and 2 m as in Trembley et al. (2009).

Samples for nutrient determination were taken at all sampling depths in the water column. Water was GF/F filtered before being stored in previously acid-washed 30 mL HDPE plastic bottles and frozen until analysis (-18°C). Analysis for inorganic nutrients (nitrite+nitrate, orthophosphate and silicate) were measured on a Smartchem200 (by AMS Alliance) autoanalyser (for more detail see Paulsen et al. (2017)). Profiles of nitrate + nitrite ($NO_x$) were completed using linear interpolation; the nitracline ($Z_{NOx}$) was determined by visual inspection of the relationship of $NO_x$ and density whereby we approximated the isopycnal and the corresponding depth at which there is a consistently increasing gradient of $NO_x$ above a 0.5 μM threshold (adapted from Omand and Mahadevan (2015)).

Chlorophyll *a* (chl *a*) concentrations were measured in triplicates at each sampling depth by filtering 250 mL of water from each sampling depth on 25mm Whatman GF/F filters (nominal pore size: 0.7 μm). Filters were then extracted in 5 mL 96% ethanol for 12–24 hours and analysed on a Turner Design Fluorometer calibrated against a chl *a* standard according to Jespersen and Christoffersen (1987). The measurements were done in triplicates. Chl *a* concentrations were used to calibrate the chlorophyll *a* fluorescence ($flu_{chl}$) profiles from the CTD. At each sampling depth for chlorophyll *a*, a factor F was



calculated as F=flu$_{chl}$/ [chl $a$]. The F-factors where then linearly interpolated between sampling depths and multiplied with flu$_{chl}$ in order to obtain a calibrated depth profile of chl $a$ from 0 to 100m. The depth of the deep chlorophyll $a$ maximum (DCM) was then calculated and compared to the DFM. The DCM and DFM were positively correlated, and after accounting for outliers, did not differ from a 1:1 relationship (p = 0.721).

At 1m and DFM depths chl $a$ was also determined on 10 µm polycarbonate filters to estimate the contribution of different size fractions to phytoplankton biomass. In order to integrate these fractions through the entire water column, we applied the same fractions to each meter from the surface and fluorescence maximum depths to the middle depth between these two and the fractions at the DFM from there to the bottom on the profile (100m), after which we summed up the contribution of each fraction for the whole water column. In the case of a third depth in between the surface and DFM, the

same process was taken between the surface and middle depth as between surface and DFM. Note that the chlorophyll $a$ from Sept. 8 (Julian day: 251) at Station 1 was only integrated to 40m due to an incomplete CTD profile.

### *2.3 Primary production*

    Primary production (PP) was measured as $^{14}$C-uptake (Nielsen, 1952) according to Markager et al., (1999). Briefly,

samples were collected at 1 m depth and at one or two additional depth with a notable DFM (26 sampling dates in total). The samples were brought to the laboratory and incubated for ca. 4 hr at *in situ* temperature in an ICES incubator (Hydro-Bios, Germany) at 11 different light intensities and in darkness. Flat 62 mL bottles (Nunc) were illuminated from both sides with white LED-light. The actual light intensity was measured before and after each incubation with a $2\pi$ sensor (Li-Cor 192UW quantum sensor) at 16 positions in the incubator. The $^{14}$C-bicarbonate (obtained from DHI, Denmark) was added to an 800

mL sample and dispensed into the Nunc bottles. In order to maximize sensitivity and save isotope, the addition of isotope was adjusted according the chl $a$ concentration and hence the expected uptake, and varied from 6 to 80 µCi per 800 mL of sample. Three Nunc bottles incubated at low, medium and high light were spiked with additional isotope, in order to measure production in three different size fractions: >10 µm, GF/F (nominal pore size: 0.7 µm) to 10 µm and < 0.7 µm. After the incubation, total production was measured as 10 mL sample taken from these three Nunc bottles and added to glass vials.

Then 500 µL 1 N HCl was added and the vials were gently bubbled 5 times over 48 hours before addition of 10 mL scintillation cocktail. The remaining ca. 52 mL in the three spiked Nunc bottles where filtered through 10 µm pore filters and the filtrate was collected. This filtrate, and the content of the other nine Nunc bottles, were filtered through GF/F filters. All filters were placed in plastic vials and acidified with 200 µL 1 N HCl. Then the vials were allowed to stand for 24 hours before they were closed and stored in a freezer. Within 1-2 months all vials counted in a Perkin Elmer TriCarb 2910 TR scintillation counter.

The $^{14}$C-uptake was calculated for each bottle and fraction (<0.7 µm, between 0.7 µm and 10 µm and >10 µm) from the effect volume and the added amount of $^{14}$C. The carbon fixation was then calculated from the DIC-concentration. PP fractions were integrated over 100m in the same way as the chl $a$ fractions.

    The areal primary production was calculated according to Lyngsgaard et al. (2014). From the carbon uptake on a

GF/F (nominal pore size: 0.7µm) filter from each bottle, the parameters in a P-I curve were estimated for each depth. These were divided with the chl $a$ concentration measured in a subsample from the same carboy from where the water for primary production was collected in order to obtain chlorophyll $a$-specific parameters for each depth. These were then extrapolated as described in Lyngsgaard et al. (2014) and multiplied with the continuous chl $a$ profile estimated from the CTD-profiles giving





volume specific P-I parameters for each depth (10 cm intervals). Finally, the daily areal production was estimated by
integrating over 24 hours for every meter down to 100m depth. Note that the primary production data from Sept. 8 (Julian
day: 251) at Station 1 was only integrated to 40m due to an incomplete CTD profile. The light intensity at each depth was
calculated from the light attenuation and the surface light measured at the nearby Zackenberg Research Station as part of the
GEM program.

### 2.4 Data analysis


Data has been divided into two seasons—summer and fall—for analyses. Summer and fall seasons correspond to the
sampling periods July 1–August 10 (Julian days 192–222) and September 4–October 6 (Julian days 247–279) respectively,
as water column properties underwent a strong transition between these two periods primarily related to the inflow of
freshwater which ceases in fall (Fig. 1f-g); see Results).


### 2.5 Environmental time series

Annual incident PAR data and sea ice break-up (Fig. 2a), as well as Zackenberg river discharge (Fig. 3c) were
obtained from the GEM program website (http://g-e-m.dk). Incident PAR ($\mu mol\ s^{-1}\ m^{-2}$; Fig. 3a), measured as part of the
GEM ClimateBasis program, is recorded every 30 minutes, using a Li-Cor quantum sensor located 2 m above terrain at the
Zackenberg Research Station, and wind velocity (Fig. 3b) is also logged as part of the ClimateBasis program. Sea ice break-
up dates, monitored by the MarinBasis program, are estimated using both satellite images and a time-laps camera situated
above the fjord at the approximate location of the MarinBasis standard sampling station (Station 3). Zackenberg river
discharge ($Q\ m\ s^{-1}$; Fig. 3c) is monitored by the GEM ClimateBasis program.

## 3 Results

### 3.1 Physical environment

In 2014, ice break up in the main fjord occurred on July 27 (Fig. 2a), thus total annual surface PAR during the open-
water was lower than the previous 9 years (Fig. 2a). Despite the late break up in 2014, the overall trend is toward an earlier
break up. Based on sea ice data from 1950-2014, ice breaks up 0.15 days year$^{-1}$ earlier corresponding to 1.2 days earlier in a
10-year period (Middelbo et al., 2019) which adds 2.6% per decade to the annual amount of PAR in the water column. There
is a 17 times difference (39.2 versus 2.36 mol m$^{-2}$ day$^{-1}$) in daily irradiance between July and October so it is clear from Fig.
2 that the date for ice break up is much more important for determining light availability for marine primary producers than
the date for sea ice formation in autumn. It is important to note however, that due to continental warming, ice broke up earlier
in the inner part of the Tyroler fjord, approximately in mid-June, based on estimates from satellite images. Thus, a square
meter of surface water in the inner part of the fjord receives almost twice (1.87) times the annual irradiance compared to the
outer fjord.

Discharge of the Zackenberg River started on June 4 (Julian day: 155), peaked on August 16 (228) with the outburst
flood from a glacial lake, reaching 169 m$^3$ s$^{-1}$, and ended on September 28 (271) in 2014 (Fig. 3c). Total accumulated discharge
of the Zackenberg River in 2014 was 0.22 km$^3$, within the normal range of annual discharge (0.13–0.34 km$^3$; Citterio et al.
2017).



CTD transects (Fig. 1d-i) offer a coarse seasonal view of the extremes in physical conditions during the ice-free
period in the fjord. Shallow salinity stratification was consistent across the horizontal gradient of the fjord even to the outer
most station in the Greenland Sea in the summer (Fig. 1f). In the fall, the upper 30 m was relatively well mixed but a pycnocline
was still present around 30 m in the fjord while, outside the fjord stratification was deeper and weaker (Fig. 1f). Turbidity
was most pronounced throughout the upper water column at the inner most stations due to run-off from the Tyroler River in
the summer time with another pronounced increase in turbidity in stations just past the outflow of the Zackenberg river (Fig.
1h). In the fall, after run-off from the rivers ceased, turbidity was lower and more homogenous throughout the upper water
column across the entire transect. An area with high turbidity was observed on the outer coast, which was likely related to
resuspension of sediment due to large ocean swells hitting the shallow area around the outer sill and the small island there
(Fig. 1i). The phytoplankton biomass, as described by chlorophyll *a* fluorescence, showed low values both at the surface and
at depth in the water column. Fluorescence was concentrated higher up in the water column in the inner most stations, but the
peak deepened moving out the fjord (Fig. 1d). However, variation in DCM during the summer in the inner fjord, does not
allow for detection of any trends across stations (Table 1; summer mean ± SD DCM at main sampling stations: $28 \pm 8.2$ m).
In the fall, florescence was more homogeneous throughout the upper water column (Fig. 1e), and the DCM moved higher up
in the water column in all stations (fall mean ± SD DCM at main sampling stations: $18 \pm 11$) except for the outer-most stations,
which were subject to deep mixing (DCM: 69 m). Though further inspection of profiles suggests that fluorescence profiles
may not be the best parameter to judge vertical distribution of biomass, rather it is best to consider the profiles of chlorophyll
*a* due to the systematic variations in fluorescence per unit of chlorophyll *a* (see Fig. S2 and results below).

In general, nutrient concentrations increased with depth. Nitrate + nitrite ($NO_x$) concentrations increased from a mean
± SD surface value of $0.15 \pm 0.27$ µM to $4.32 \pm 1.21$ µM at 100 m depth. Phosphate concentrations increased slightly from
$0.38 \pm 0.30$ µM at the surface to $0.80 \pm 0.35$ µM at 100 m depth, though this increase is not significant indicating that
phosphorous in not used up at the surface and thus not limiting. Indeed, the average $NO_x$ to phosphate ratio for the data set is
$2.0 \pm 2.5$ (mean ± SD), much below the Redfield value of 16, as such, communities are limited by nitrogen with phosphorous
in surplus. Silicate increased from $4.35 \pm 1.88$ µM to $6.87 \pm 0.75$ µM at 100 m depth, though at the surface silicate
concentrations are variable and range from 0.69 to 10.0 µM, likely due to high concentrations of silicate in the meltwater run-
off (Paulsen et al. 2017). For details of nutrient profiles see Fig. S1 in Paulsen et al. (2017). The average nitracline ($Z_{NOx}$) in
the data set is $29 \pm 9.5$ m (mean ± SD), and not different among stations or between seasons (t-test: $p = 0.07$). However, there
was a tendency for a shallower $Z_{NOx}$ in the summer, especially in the innermost Station 1. Due to high turbidity in Station 1
in the summer, biomass was concentrated higher up in the water column and thus had not depleted nitrate down as far (Table
1).

The mixed layer depth ($Z_m$) increased over the season at all stations (Fig. 4a; Table 1) from a mean (± SD)
of $9.3 \pm 4.6$ m in the summer to $27.1 \pm 5.5$ m in the fall excluding the last sampling date of Station 4 which had an unusually
deep $Z_m$ (86m). Stratification index (SI) however was not significantly different between seasons (mean ± SD SI in summer
and fall respectively: $6.3 \pm 3.5$ and $3.4 \pm 0.9$ kg m$^{-3}$).



Photic depth ($Z_p$) at Station 1 increased steadily (from 16.1 to 38.3 m) over the season while in Station 2, $Z_p$ increased rapidly in the beginning of the season and then only slightly toward the end (Fig. 4b; Table 1). There was no change in $Z_p$ in Station 3 throughout the season (mean ± SD: 26.5 ± 2.5 m), while photic depth in Station 4 had the opposite pattern of Station 1, decreasing over the growing season (from 38.4 to 18.9 m; Fig. 4b; Table 1), though this trend may be confounded by a

resuspension event related to large ocean swells late in the season. As expected, there was a negative relationship of photic depth with average surface (0- 5 m) turbidity ($Z_p = 24.4\ e^{surf.\ turb.\ *\ -3.99}$; $R^2 = 0.48$; $p < 0.0001$) as in Murray et al. (2015).

The ratio of $Z_p$: $Z_m$ indicates if a productive DCM is possible as it requires enough light is available below the pycnocline. The trend was similar among stations (Fig. 4c) with increasing values from ice break up reaching a maximum at

the very end of July and beginning of August (Julian days 208-222) and then decreased to values of one or less beginning in early September (Julian day 247) as the photic and mixing depths met. Thus, in most cases, there was sufficient light below the pycnocline to allow for a productive DCM. Similarly, the ratio of the photic depth to the nitracline ($Z_p$: $Z_{NOx}$) was 1.1 ± 0.6 (mean ± SD) throughout the season, indicating that across seasons the depth of the nitracline was driven by light and thus phytoplankton uptake of nitrate, again allowing for a productive DCM. On the other hand, the ratio of $Z_m$: $Z_{NOx}$, which can

indicate the potential of nutrient replenishment in the mixed layer, was significantly different between seasons (t-test: $p <$ 0.001) with a mean of 0.37 ± 0.17 in the summer and 0.99 ± 0.4 in the fall; thus, implying that the mixing depth in the summer was much shallower than the depth were nutrients are available, but that in the fall mixing depths were sufficient to bring nutrients into the mixed layer.


### 3.2  Chlorophyll a and primary production

Water column chl *a* varied among stations and along the season (Fig. 5a; Table 1), with the highest integrated values found at the outermost Stations 3 and 4 (mean ± SD: 34.2 ± 16.1 mg chl *a* m⁻²). Values at Stations 1 and 2 were significantly lower (mean ± SD: 17.4 ± 6.9 mg chl *a* m⁻²; paired t-test; p = 0.002) throughout the summer and fall months. At Stations 2, 3

and 4 there was a trend of a majority of large cell size earlier in the season, whereas later in the season small cells dominated. At Station 1, small cells dominated throughout the season (Fig. 5a).

The average areal primary production in the study was 92 mg C m⁻² day⁻¹ ranging from 10.6 to 628 mg C m⁻² day⁻¹ (Table 1; Fig. 5b). However, areal production showed little pattern seasonally or spatially, apart from evidence of an under-

ice bloom at Station 3 on July 11ᵗʰ (Julian day 192) which reached 628 mg C m⁻² day⁻¹ (Table 1). Other notable features are the high primary production on the first sampling date in Station 1 (207 mg C m⁻² day⁻¹; Table 1) despite low chlorophyll *a* biomass (12.5 mg chl *a* m⁻²), whereas Station 4 showed a small peak in primary productivity in late summer (155 mg C m⁻² day⁻¹; Table 1) in accordance with the peak in chlorophyll *a* biomass. Only 6 of the 26 primary production estimates were above 100 mg C m⁻² day⁻¹, 5 of which were before the 10th of August (Julian day: 222), which suggests a tendency toward a

higher production over the first approximately four weeks of the ice-free period. However, there was no difference observed between areal primary production in summer and fall (summer mean ± SD: 83.8 ± 57.3 mg C m⁻² day⁻¹—excluding under ice bloom; fall: 60.4 ± 27.3 mg C m⁻² day⁻¹; t-test: p > 0.05). There is some indication in stations 1, 2 and 3 for a decrease in primary production in the summer season and recovery to similar rates in the fall, while station 4 shows an opposite trend peaking in late summer and falling back off in the fall. But, low sample size prohibits statistical analysis of these trends.



Similarly, it is difficult to observe consistent patterns of fractioned primary production observed among stations or along the season (Fig. 5b).

While patterns in areal primary production spatially and over the season are difficult to discern, there are some recognizable patterns when looking at the depth distribution of chlorophyll $a$ and primary production (Fig. S2 and S3

respectively). These patterns are summarized in Fig. 6 where we distinguish between inner and outer fjord patterns. Inner fjord stations such as Station 1 and 2 experienced high turbidity (Table 1) and hence greater light attenuation in the surface. In these stations carbon fixation was confined to the upper 5 meters. Indeed, > than 50% of the total water column primary production took place above 4 meters in the station 1 in the summer. On the other hand, Stations 3 and 4 exhibit maximum production at depth and 50% of areal production is reached further down around 20 meters in the summer months. We

observed opposite patterns in the fall however; production is concentrated further down around 12- 15 m in the inner fjord stations compared to in the summer, while production moves further up in the water column in the outer fjord stations in the fall (Fig. S2).

### 3.3  Photosynthetic parameters

The chlorophyll $a$ standardized maximum carbon uptake ($P^B_m$) varied from 0.072 to 2.62 g C g$^{-1}$ chl $a$ h$^{-1}$ with a mean (± SD) value of 0.66 ± 0.56 g C g$^{-1}$ chl $a$ h$^{-1}$. High values, above 2 g C g$^{-1}$ chl $a$ h$^{-1}$, where observed only in the beginning of the growing season (i.e. before the end of July; Table S1). Values from 1-meter depth were higher than the values from the DFM for 18 out of 20 days, but the difference was not significant (paired t-test, p=0.19). The light utilization efficiency value ($\alpha_B$) varied from 0.67 to 48 g C g$^{-1}$ chl $a$ mol$^{-1}$ photons m$^2$ with a mean (± SD) value of 8.72 ± 8.52 g C g$^{-1}$ chl $a$ mol$^{-1}$ photons

m$^2$ and no difference was detected between depths (Table S1). The Ik-values ($P^B_m / \alpha_B$) express the light level that is saturating for carbon fixation. Values were low ranging from 5.8 to 67 µmol photons m$^{-2}$ s$^{-1}$ with a mean (± SD) value of 26 ± 15 µmol photons m$^{-2}$ s$^{-1}$. The values at 1m were slightly higher, 29 versus 26 µmol photons m$^{-2}$ s$^{-1}$, but the difference was not significant. Some high values were seen from mid-July to mid-August at 1 m, but overall there was no detectable pattern in the values (Table S1).


### 4  Discussion

### 4.1  Areal patterns of primary production indicate low light adapted communities

This study contributes significantly to the sparse knowledge of spatial and temporal patterns of primary productivity

in Northeastern Greenland fjord systems, and expands on previous studies of primary productivity in Young Sound confirming that it is indeed a low productivity fjord throughout the open water season. Primary production rates in this data set the fall within the range of values measured previously in Young Sound by Rysgaard et al. (1999) (This data set: 206.8 – 10.8 mg C m$^{-2}$ day$^{-1}$ (Fig. 5b; Table 1); Rysgaard et al. (1999): 277.9 – 4.2 mg C m$^{-2}$ day$^{-1}$) with the exception of the high rates of primary production measured under the ice at Station 3 on July 11. However, these rates are much lower compared to other Arctic

fjords (Simo-Matchim et al. (2016)—compiled literature review within).

Low productivity in Young Sound was initially considered to be a consequence of the late break-up of sea ice thus resulting in a short productive season (Nielsen et al., 2007; Rysgaard et al., 1999). Earlier studies also showed generally low



rates of primary production under the ice in Young Sound (28 – 122.5 mg C m$^{-2}$ day$^{-1}$) due to thick snow cover on ice creating

poor light conditions (Glud et al., 2007; Nielsen et al., 2007; Rysgaard et al., 2001). Our study however reports an under-ice primary production rate of 628 mg C m$^{-2}$ day$^{-1}$ (Fig. 5b; Table 1).These high rates of primary production are short lived however and correspond to the days just before the sea ice breaks up when ice has thinned, snow cover has melted away and more light can reach the water column (Glud et al., 2007; Rysgaard et al., 2001). This peak in primary production likely consumes much of the available nutrients; indeed, NO$_x$ concentrations between 1 – 30 m were less than 1 μM during this

peak. Nielsen et al. (2007) also report low NO$_x$ concentrations below the ice already in June (< 2 μM in the upper 30 m of the water column).

Due to general low productivity, low temporal resolution, and high variability in the data, it is difficult to discern any consistent seasonal patterns of primary production in the stations sampled (Fig. 5b) in spite of very clear environmental

changes, most notably in the influence from run-off from land changing stratification patterns and the change in day length. However interesting, is that steady rates of primary productivity are documented well into the fall, and while the rates are still low, they are comparable to that of the summer season (Table 1). In Young Sound, we do not notice a traditional spike in primary productivity in the late summer or fall—often termed a "fall bloom" and typical in many high latitude systems (Wassmann and Reigstad, 2011)—rather primary productivity in September thru early October remain steady and rates are

not different than those measured during July and August, even though daily PAR in the fall is less than a quarter of the summer PAR due to shorter day lengths (Fig. 3a). In Godthåbsfjord in West Greenland, when daily PAR decreases to a quarter of the summer PAR in November, primary production rates also decrease to less than a quarter of summer rates (Juul-Pedersen et al., 2015). Simo-Matchim et al. (2016), also report generally lower rates of primary productivity in Arctic fjords in the late fall, thus making our finding significant. This could indicate that phytoplankton in Young Sound are so well adapted to low

light conditions that it allows for a low but steady rate of primary productivity well throughout the fall when some nutrients are able to be mixed into the photic zone.

In this study, we measured primary production using P-I curves (Table S1), which gives some additional indications of photosynthetic performance which we can use to compare across systems. The I$_k$ parameter, or the light intensity at which

photosynthesis is initially saturated, ranged from 6–67 (mean ± SD: 26 ± 15) μmol photons m$^{-2}$ s$^{-1}$, which is rather low compared to other studies carried out in the Arctic (Gallegos et al. 1983; Fernández-Méndez et al. 2015; Jenson et al. 1999; Simo-Matchim, personal communication). Values from those studies range from 18.9–533 μmol photons m$^{-2}$ s$^{-1}$, with an average of 97.1 μmol photons m$^{-2}$ s$^{-1}$, an order of magnitude higher than the values we find in this study. Even in the central Arctic Ocean where ice cover heavily limits light availability, I$_k$ values average 293 in August and 15 μmol photons m$^{-2}$ s$^{-1}$ in

September (Fernández-Méndez et al., 2015). River influenced Labrador fjords (Simo-Matchim et al. 2016) also show higher values of 78.6–203 μmol photons m$^{-2}$ s$^{-1}$ (Simo-Matchim, personal communication). The low I$_k$ values we find in this study are the main evidence that we have to argue that plankton communities in Young Sound are especially adapted to low light conditions, thus giving them an advantage during summer when turbidity is high and during fall with so little incident irradiance.


**4.2 *Freshwater input determines vertical patterns in primary production***





Light is limiting in Young Sound during the spring and early summer due to the presence of sea ice, but light limitation during the summer after the ice breaks up can be linked to the turbidity of the water column induced by meltwater run-off (Murray et al., 2015), which gradually increases throughout the summer and ceases in the fall (Fig. 3c). This affect was most noticeable at Station 1, which is strongly affected by run-off from the Tyrol River, where the photic depth deepened from 16 m in the summer to 38 m in the fall (Fig. 4b). In the fall however, light is limited due to decrease in daily PAR; and sun angles are much lower than in the summer. Fig. 6 illustrates this difference, where PAR is attenuated rapidly at the surface in the summer due to the turbid surface layer in the inner fjord, whereas in the fall, surface PAR starts off lower, but is attenuated slower when there is less turbidity in the upper water column. A part from some variation, the nitracline however remained very consistent across stations and seasons at approximately 30 m depth, just below the average photic depth for all stations (Fig. 6).

The effects of this changing light environment are seen in the distribution of biomass and carbon fixation in the water column. In the outer fjord during the summer, primary productivity peaks closer to the nitracline exhibiting a classic DCM as would be expected in an oligotrophic system, whereas during the fall productivity moves higher up in the water column due to short day length. However, in the turbid inner fjord during the summer, the production takes place higher up in the water column as is the case in a typical estuary, however during the fall the production actually moves to an intermediate depth (12–15m) in the water column. Hence, the distribution of biomass and productivity in the water column is indicative of a dual light and nutrient limited system. Phytoplankton face a constant trade off: in summer, phytoplankton can still grow—though with low rates—in the deeper layers bordering the nitracline. However, in the fall light is so low and days short forcing phytoplankton closer to the surface where light limitation is less pronounced but nutrients are strongly limiting, unless they are adapted to low light in which case they can afford to be further down in the water column and closer to nutrients.

### 4.3 Nutrient availability is dependent on circulation inside and outside the fjord

Nitrate availability in the photic zone is determined both by winter surface concentrations prior the spring bloom and vertical mixing processes that replenish nitrate to the photic zone during the summer and autumn. In Young Sound, $NO_x$ concentrations in February in surface water are around 3 μM (Rysgaard et al., 1999). Outside the fjords on the East Greenland Shelf in April, nitrate concentrations are only 3-5 μM (Michel et al., 2015), and maximum concentrations in the summer inside and outside the fjord at depth are around 4-8 μM (this study, Paulsen et al. 2017). As a comparison, $NO_x$ concentrations during winter in the surface waters of Godthåbsfjord (West Greenland) reach up to 12.5 μM (Juul-Pedersen et al., 2015), while concentrations just below the photic zone can be around 10 μM in the Disko Bay (Sejr et al., 2007). So, the location of Young Sound on an "outflow" shelf dominated by surface water already depleted in nutrients (Michel et al. 2015) is clearly an important component contributing to limiting nutrients within the fjord.

Lack of mixing processes during summer also have limited capacity to replenish nitrate to the photic zone in Young Sound. First, the maximum tidal amplitude in Young Sound is only 0.8 m (Bendtsen et al., 2007), which is low compared to the tidal amplitude in the more productive Godthåbsfjord of up to 5 m (Blicher et al., 2013). Additionally, in Young Sound, river discharge throughout the summer creates a surface lens of freshwater and hence mixed layer depth shallower than 10m, but in the fall when river run-off ceases, the mixed layer depth deepens to approximately 30m (Fig. 6) potentially bringing up some nutrients—the nitracline also sits around 30m depth—which may account for the low but significant rates of primary



production during that time period. Though patterns of stratification are very different between seasons the water column remains stratified well into the fall. Although there was some increase in wind speed, as well as a large storm towards the end of the season (Fig. 3b) and the stratification index in fall tended to be lower than in the summer (Table 1), there was no difference in stratification index between seasons. Furthermore, SI values in this study are 2-3 times higher than SI values

calculated in the same way in Labrador fjords also influenced by river run-off (Simo-Matchim et al., 2016). This suggests that any increase in the stratification index due to freshening could reduce vertical mixing even further.

### *4.4   Perspectives*

Ice break up occurs much later in Young Sound than in other well-studied West Greenland fjords (e.g. Godthåbsfjord;

Disko Bay) limiting light availability in the water column which is why it has been suggested that an increased length of open water season would benefit primary production in Young Sound (Glud et al., 2007; Rysgaard et al., 1999). However, current trends in ice break-up dates show that sea ice is breaking up 1.2 days per decade in Young Sound, amounting to a less than 1% addition of surface PAR to the annual open water PAR budget. Later ice formation in the fall will also not likely increase the annual PAR as the incident irradiance in the fall is already low, however, enhanced wind mixing from storms due to later

ice formation could increase the re-supply nutrients into the photic zone in the fall. Though it is still unlikely that changing ice conditions will be a strong driver for change in the future, even if there is an increase in the open water season. With ongoing climate change, we expect an increase and earlier onset of run-off causing the turbidity and the strong stratification that we see in this study. Mernild et al. (2008) model up to 5 times increase in discharge from the Zackenberg drainage basin in 50 years time, which is likely to influence the freshwater content of the fjord and coastal waters.


On the other hand, there is no evidence yet of increased discharge in Young Sound, and there has been no change in salinity in the upper 30 m of the water column over the last 13 years (Sejr et al., 2017), as the estuarine circulation in Young Sound results in a short residence time of ~1 month for fresh surface waters inside the fjord (Bendtsen et al., 2014). Instead, Sejr et al. (2017) report increased freshening of the 30-50 m layer of the water column inside the fjord over the last 13 years,

which they attribute to exchange of freshening shelf waters inside the fjords. It is speculated that shelf waters are freshening due to the accumulation of run-off from the numerous fjords along the Northeast coast of Greenland in the East Greenland Current (Bendtsen et al., 2014; Sejr et al., 2017) or changes could be related to melting of sea ice during summer (Boone et al., 2018). The import of the freshened waters however will likely not provide the system with any extra nutrients, as East Greenland coastal waters are rather nutrient deplete (Michel et al., 2015). Furthermore, there is little exchange in Young

Sound in water deeper than 50 m, due to the shallow entrance sill (45 m; Bendtsen et al. (2007)), though there is a decreasing trend in salinity in bottom water inside the fjord, likely due to the increasing freshwater content inside the fjord (Sejr et al., 2017) which is mixed down to deeper depths via turbulent diffusion (Bendtsen et al., 2007).

Freshening in other parts of the Arctic ocean has caused a deepening halocline resulting in a deepening nitracline and

chlorophyll *a* maximum (McLaughlin and Carmack, 2010). It has been modeled that increased run-off in Young Sound would not necessarily influence the mixed layer depth (Bendtsen et al., 2014). However, it would decrease the surface salinity and increase the freshwater content, thereby increasing the density difference between the upper and lower water column—increasing the stratification index and the amount of energy required for mixing and hence nutrient replenishment to surface layers in the fall. Studies from a tidewater glacier fjord (Godthåbsfjord) in Greenland report that meltwater, while





it creates strong stratification, actually enhances primary productivity due to the upwelling driven by sub-glacial discharge bringing-up nutrients throughout the summer melt period. Stratification induced by the meltwater actually stabilizes the water column replete with nutrients sustaining high primary productivity throughout the summer (Juul-Pedersen et al., 2015; Meire et al., 2017). Consequently, a mechanism has been suggested whereby fjords receiving run-off from rivers or land terminating glaciers lack the mixing and nutrient replenishment provided by tidewater glaciers, resulting in lower

productivity fjords (Meire et al., 2017). On the other hand, primary production rates in river-influenced Labrador fjords are similar to those found in Godthåbsfjord despite their being ice covered throughout the spring (Simo-Matchim et al., 2016). Therefore, it is difficult to make generalizations about glacial fjords across the Arctic. It is likely an interplay between ice cover and timing of ice break up, freshwater input (either locally produced or allochthonous to the system), degree of stratification and mixing properties (including tidal mixing which can vary dramatically around the coast of Greenland),

coastal boundary currents and depth of entrance sills determining exchange of water masses within a fjord, as well as glacier type (marine vs. land-terminating) that determine the overall circulation and productivity of a fjord.

## 5  Conclusions

Seasonal observations from this high-Arctic fjord show a system that is characterized by an isolated surface layer due to run-off from land-terminating glaciers that exhibits a very shallow mixed layer in the summer and a deeper mixed layer in the fall (Fig. 6). There is a spatial gradient moving out the fjord whereby light is attenuated rapidly in the inner fjord during the summer due to turbidity introduced by rivers (Fig. 6a); a majority of primary production takes place in the upper meters. In the fall, a shallow DCM forms in the inner fjord despite a low light environment, likely due to low light adaptivity of the

phytoplankton. On the other hand, the outer fjord exhibits a more traditional DCM in the summer where there is sufficient light for phytoplankton to grow as close to the nitracline as possible—nitrate is depleted down to 30m throughout the growing season. In the fall the DCM moves to the surface away from the nitracline in the outer fjord, due to low sun angle and limited light availability. While an extremely unproductive fjord, minimal productivity is maintained in Young Sound throughout the summer and the fall through an interplay and trade-off between a) low light availability, which in the spring is caused by the

presence of sea ice, in the summer by river induced turbidity, and in the fall due to short day length and low sun angle, and b) low nutrient availability, due to the inherently low nutrient concentrations and depletion of nutrients early on in the season under the ice combined with intense stratification throughout the season that allows for little vertical mixing of the water column. Thus, we conclude that future productivity in Young Sound will likely be more affected in the future by increased run-off locally and freshening of the coastal current from land rather than the length of the open water season.

Currently there are few seasonal primary production studies in glacier influenced fjords across the Arctic and even fewer around Greenland (Simo-Matchim et al., 2016). More studies are needed to determine the main processes controlling productivity in different types of fjords—e.g. those influenced by marine vs. land-terminating glaciers, silled vs. non-silled and shallow silled fjords, ice covered vs. non-ice-covered fjords, as well as fjords influenced by different boundary currents or mixing processes— before large generalizations about the future productivity of Greenland fjords can be made.


**Author contributions**





JMH analyzed data and wrote the manuscript. TJP and SM measured primary production. SM modelled light and primary production data and was involved in data analysis and interpretation along with MKS and MLP. SM, TJP, LM and MKS
designed the study and EFM, MLP, and MKS collected complementary data in the field. ALL AUTHORS significantly contributed to the final interpretation of the results, including commenting and editing of the manuscript.

**Acknowledgments**

This study is a contribution to the MarineBasis-Zackenberg programme, part of the Greenland Ecosystem Monitoring programme (GEM), and the Arctic Science Partnership (ASP). This study was supported by the Danish Environmental Protection Agency's program for Arctic research (DANCEA), The Carlsberg Foundation, the Horizon 2020 project INTAROS funded by the European Union, and the project MicroPolar (RCN 225956) funded by the Norwegian Research Council. Parts of the data included in this study were provided by the ClimateBasis, MarineBasis, and GeoBasis programmes
of the Greenland Ecosystem Monitoring Programme. JMH is supported by European Union's Horizon 2020 research and innovation programme under the Marie Sklodowska-Curie grant agreement No. 752325, whereby the research contained in this document reflects only the authors' views and the Agency is not responsible for any use that may be made of the information it contains. Finally, we would like to acknowledge Egon Frandsen, Kunuk Lennert, and Ivali Lennert for excellent assistance during field work.


**Competing interests**

The authors declare that they have no conflicts of interest

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



**Table 1**. Summary table. $Z_p$ = photic depth (m), Turb $Z_0$ = surface turbidity (FTU), $I_0$ = surface irradiance (μmol photon m$^{-2}$ s$^{-1}$), $Z_m$ = mixed layer depth (m), SI = stratification index, $Z_{NOx}$ = nitracline depth (m), DCM = depth of chlorophyll *a* maximum (m), Chl = areal chlorophyll *a* (mg chl m$^{-2}$), PP = areal primary production (mg C m$^{-2}$ day$^{-1}$).

| Date | Julian day | Season | $Z_p$ | Turb $Z_0$ | $I_0$ | $Z_{comp}$ | $Z_m$ | SI | $Z_{NOx}$ | DCM | Chl | PP |
|---|---|---|---|---|---|---|---|---|---|---|---|---|
| **Station 1** | | | | | | | | | | | | |
| 21-Jul | 202 | Summer | 16.1 | 2.01 | 44.95 | 27 | 7 | 4.02 | 19 | 25 | 12.45 | 206.75 |
| 1-Aug | 213 | Summer | 22.7 | 1.70 | 16.47 | 28 | 6 | 11.17 | 18 | 10 | 4.97 | 25.12 |
| 8-Sep | 251 | Fall | 27.2 | 0.26 | 16.99 | 32 | 26 | 4.25 | 34 | 24 | 17.32 | 49.21 |
| 20-Sep | 263 | Fall | 34.2 | 0.23 | 8.12 | 34 | 25 | 3.48 | 22 | 10 | 27.59 | 105.50 |
| 27-Sep | 270 | Fall | 38.3 | 0.15 | 3.41 | 32 | 27 | 3.48 | 35 | 21 | 23.33 | 73.23 |
| **Station 2** | | | | | | | | | | | | |
| 17-Jul | 198 | Summer | 12.9 | 2.49 | 23.84 | 24 | 7 | 9.36 | 36 | 41 | 16.52 | 66.47 |
| 27-Jul | 208 | Summer | 25.3 | 2.72 | 41.30 | 40 | 7 | 12.96 | 34 | 28 | 14.81 | 102.90 |
| 5-Aug | 217 | Summer | 26 | 0.69 | 33.96 | 41 | 7 | 7.54 | 22 | 26 | 8.48 | 37.05 |
| 6-Sep | 249 | Fall | 24 | 0.35 | 17.19 | 28 | 22 | 3.71 | 41 | 10 | 21.90 | 77.46 |
| 13-Sep | 256 | Fall | 30.5 | 0.29 | 14.37 | 35 | 32 | 3.77 | 49 | 17 | 23.22 | 66.15 |
| 27-Sep | 270 | Fall | 27.7 | 0.25 | 3.14 | 23 | 28 | 3.41 | 17 | 6 | 21.14 | 26.11 |
| **Station 3** | | | | | | | | | | | | |
| 11-Jul | 192 | Summer | 25.6 | 8.34 | 11.07 | 39 | 19 | NA | 27 | 30 | 51.72 | 627.52 |
| 19-Jul | 200 | Summer | 26.2 | 0.79 | 48.67 | 39 | 9 | 5.84 | 15 | 33 | 46.97 | 112.72 |
| 30-Jul | 211 | Summer | 30.2 | 0.21 | 19.00 | 38 | 6 | 2.92 | 31 | 31 | 47.54 | 54.75 |
| 7-Aug | 219 | Summer | 25.9 | 0.40 | 20.58 | 35 | 6 | 6.18 | 21 | 21 | 30.79 | 39.19 |
| 4-Sep | 247 | Fall | 21.3 | 0.53 | 18.17 | 26 | 21 | 3.95 | 30 | 9 | 20.19 | 53.23 |
| 16-Sep | 259 | Fall | 26.1 | 0.34 | 9.49 | 27 | 22 | 3.62 | 16 | 14 | 24.40 | 86.23 |
| 28-Sep | 271 | Fall | 28.4 | 0.54 | 7.19 | 28 | 23 | 3.38 | 27 | 6 | 15.57 | 96.87 |
| 4-Oct | 277 | Fall | 27.6 | 0.40 | 3.72 | 24 | 41 | 2.91 | 44 | 38 | 29.51 | 74.41 |
| 6-Oct | 279 | Fall | 27.6 | 0.40 | 2.79 | 22 | 32 | 2.93 | NA | 38 | 29.45 | 52.17 |
| **Station 4** | | | | | | | | | | | | |
| 24-Jul | 205 | Summer | 38.4 | 0.09 | 33.54 | 49 | 14 | 2.58 | 34 | 40 | 16.83 | 32.58 |
| 3-Aug | 215 | Summer | 38.4 | 0.13 | 22.64 | 43 | 17 | 3.31 | 31 | 30 | 36.03 | 88.46 |
| 10-Aug | 222 | Summer | 30.8 | 0.08 | 32.65 | 42 | 7 | 3.60 | 23 | 25 | 75.50 | 155.46 |
| 11-Sep | 254 | Fall | 32.8 | 0.52 | 14.62 | 38 | 24 | 4.27 | 32 | 30 | 26.51 | 34.55 |
| 18-Sep | 261 | Fall | 31.4 | 0.18 | 12.20 | 35 | 29 | 3.90 | 33 | 20 | 20.79 | 10.75 |
| 2-Oct | 275 | Fall | 18.9 | 0.75 | 1.78 | 11 | 86 | 0.68 | 45 | 10 | 40.93 | 40.37 |




**Fig. 1.** Satellite imagery of Young Sound, Northeast, Greenland (a-c). Left panel shows the location of all CTD sampling stations (small dots) and the 4 main sampling stations (large dots; a), and the right panels show the actual ice and snow cover on July 12, 2014 (top, b) and October 11, 2014 (bottom; c). Fluorescence (d, e) salinity (f, g), and turbidity (h, i) 710 contour plots of the CTD transects in summer (left, August 8, 2014) and fall (right, October 4, 2014).

**Fig. 2.** Average (2004-2014) daily surface PAR (black line curve) in Young Sound over one year (a). Black horizontal bars show ice cover for the years 2004-2014 (a). Actual PAR per year during the ice-free season (b). PAR data are taken from the Greenland Ecosystem Monitoring (GEM) database, and ice cover is estimated from daily photos taken from a camera situated 715 on land approximately looking down on Station 3 (*n.b.* ice break up at other main stations likely occurred on different dates)

**Fig. 3.** Daily PAR (mol m$^{-2}$ day$^{-1}$; a), average wind velocity at 10 min intervals (m s$^{-1}$; b), Zackenberg River discharge (Q m$^3$ s$^{-1}$; c) and accumulated discharge (km$^3$; red line; c).

**Fig. 4.** Mixed layer depth ($Z_m$; a), photic depth ($Z_p$; b), and the ratio between the two ($Z_p$: $Z_m$) over the growing season at each station.

**Fig. 5.** Integrated chlorophyll *a* (mg C m$^{-2}$; a) and primary production (mg C m$^{-2}$ day$^{-1}$; b) for each fraction at each station over the growing season. Note: Data from Station 1 on 8-Sep only integrated to 40m.


**Fig. 6.** Conceptual diagram of the water column profiles for density, PAR, nitrate + nitrite ([NO$_x$]) and chlorophyll *a* in the inner fjord (a) and outer fjord (b) during summer and fall in Young Sound, where $Z_m$ is the mixed layer depth, $Z_p$ is the photic depth, and $Z_{NOx}$ is the nitracline.







**Fig. 1.**

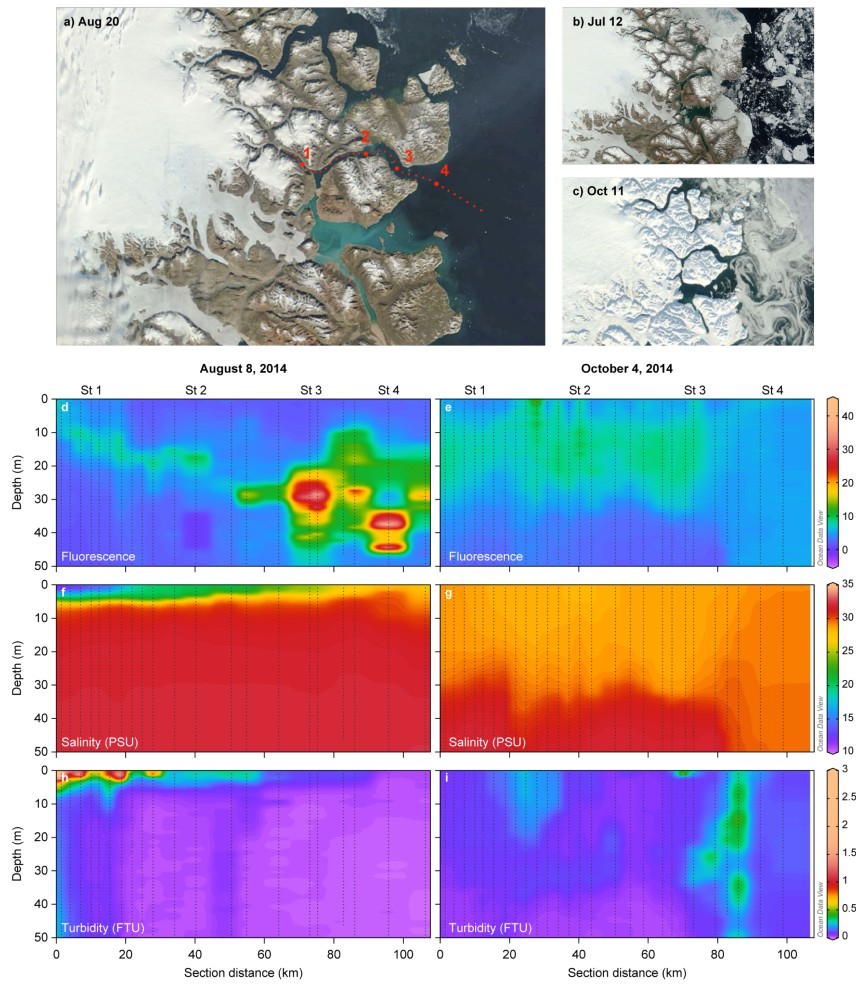





**Fig. 2.**


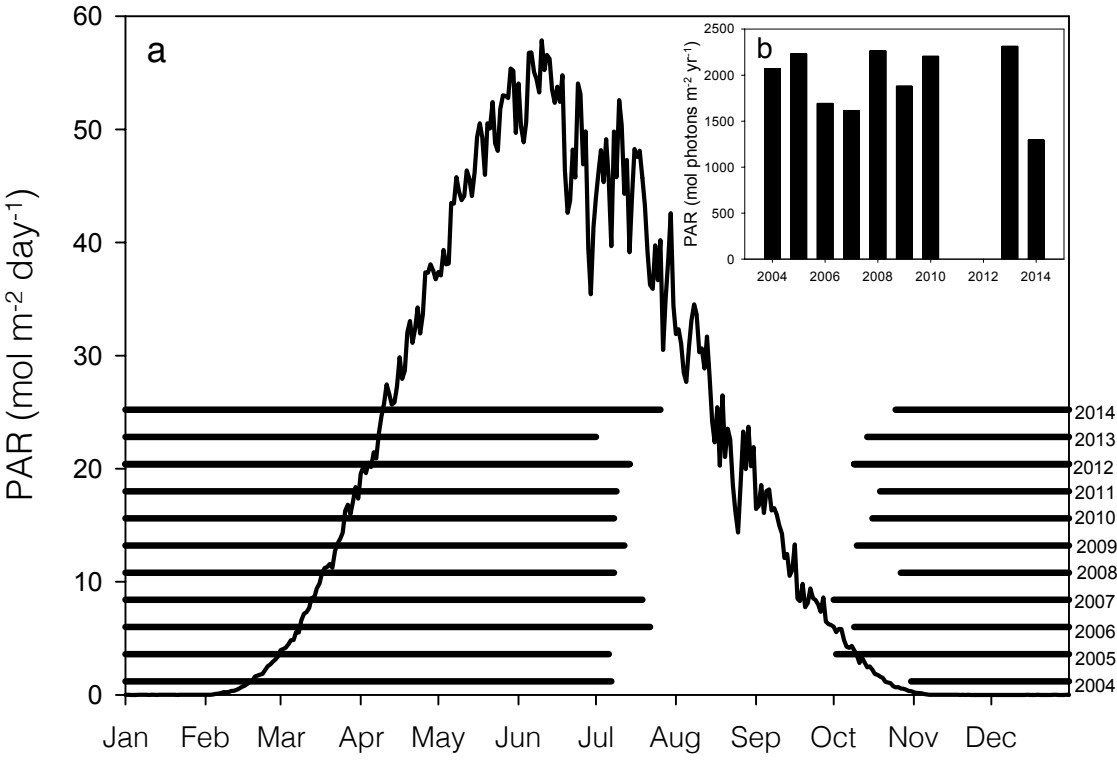





**Fig. 3.**

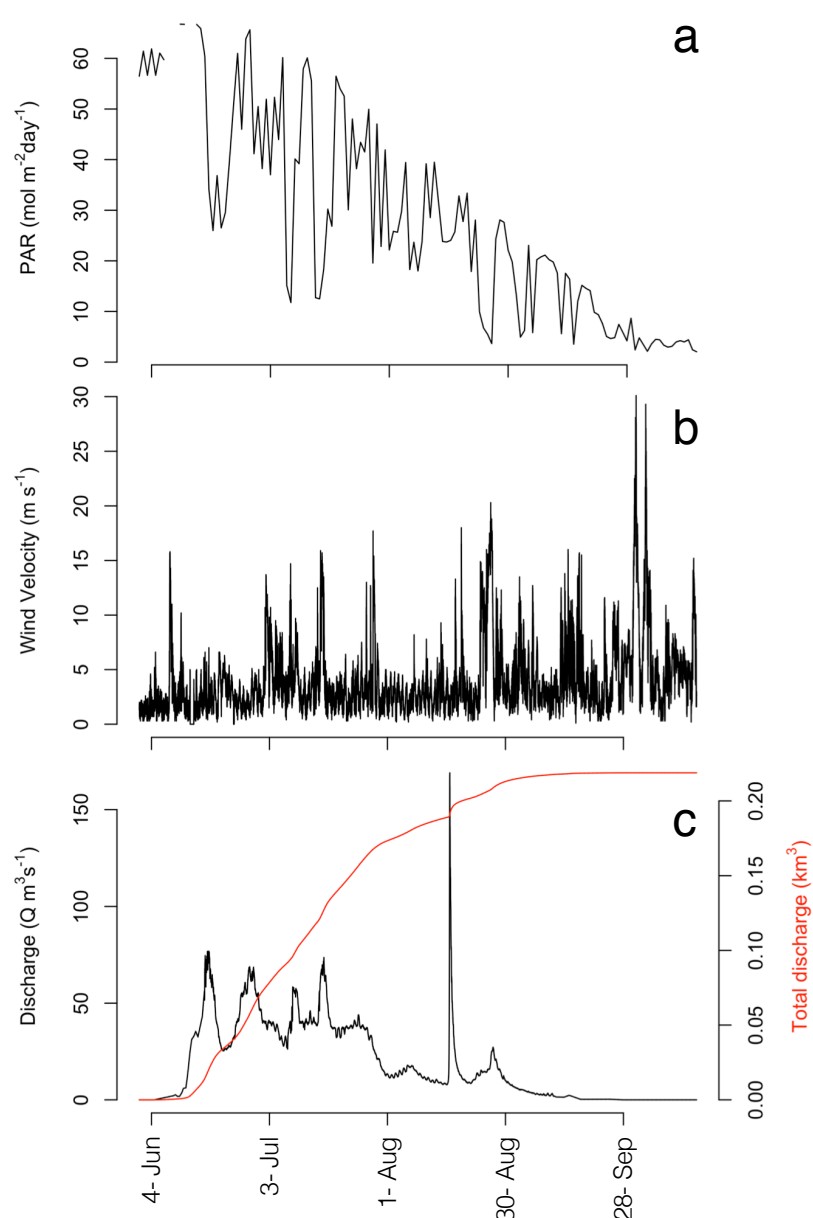



**Fig. 4.**

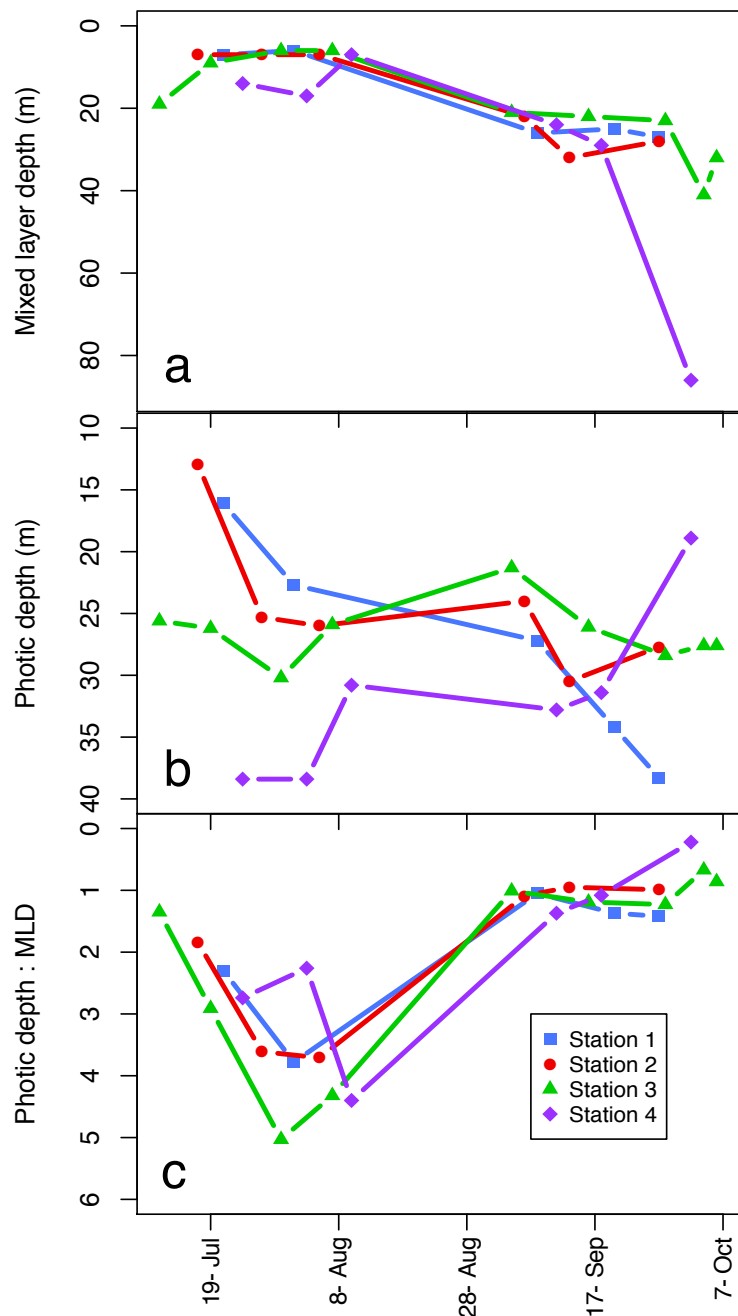





**Fig. 5.**

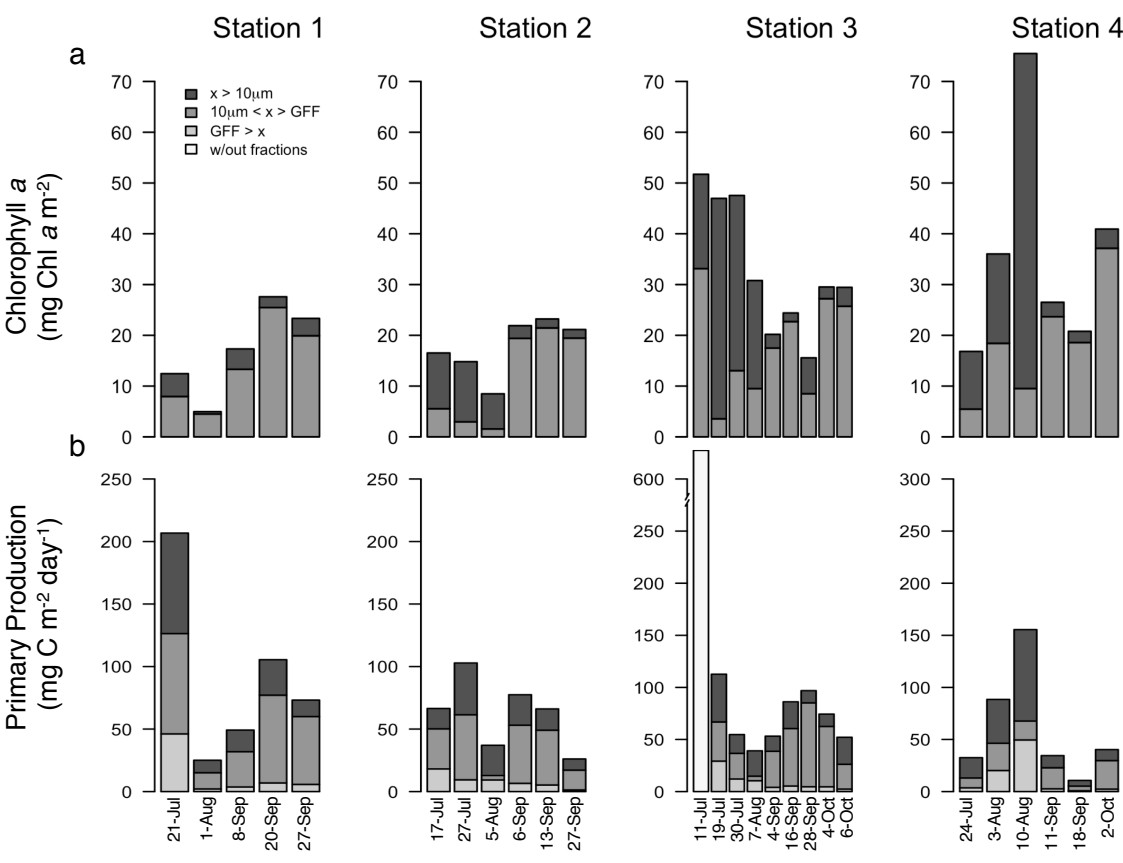






**Fig. 6.**

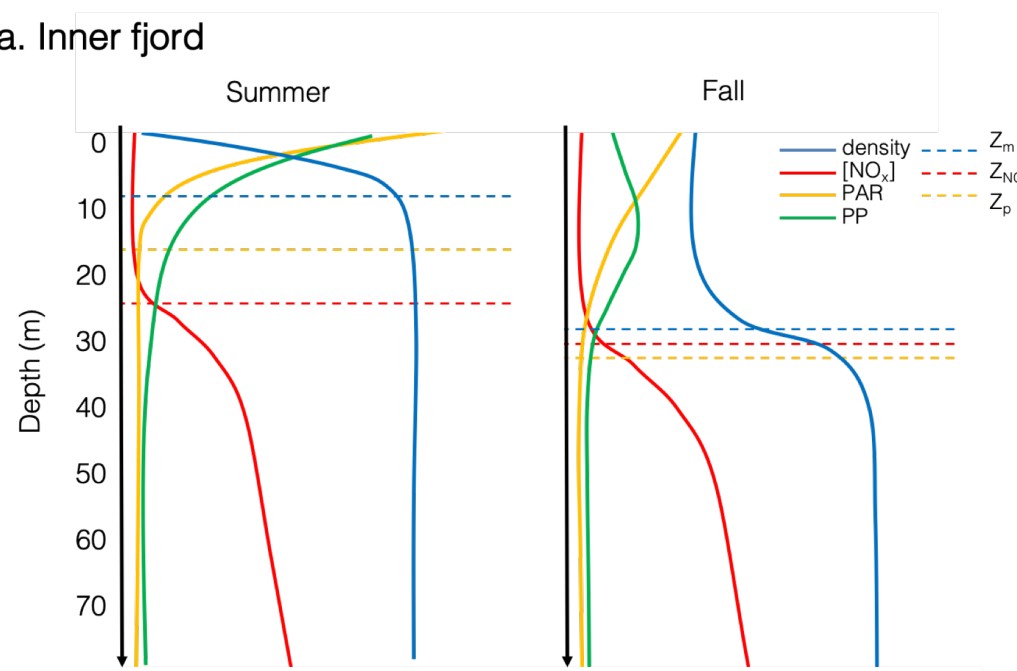

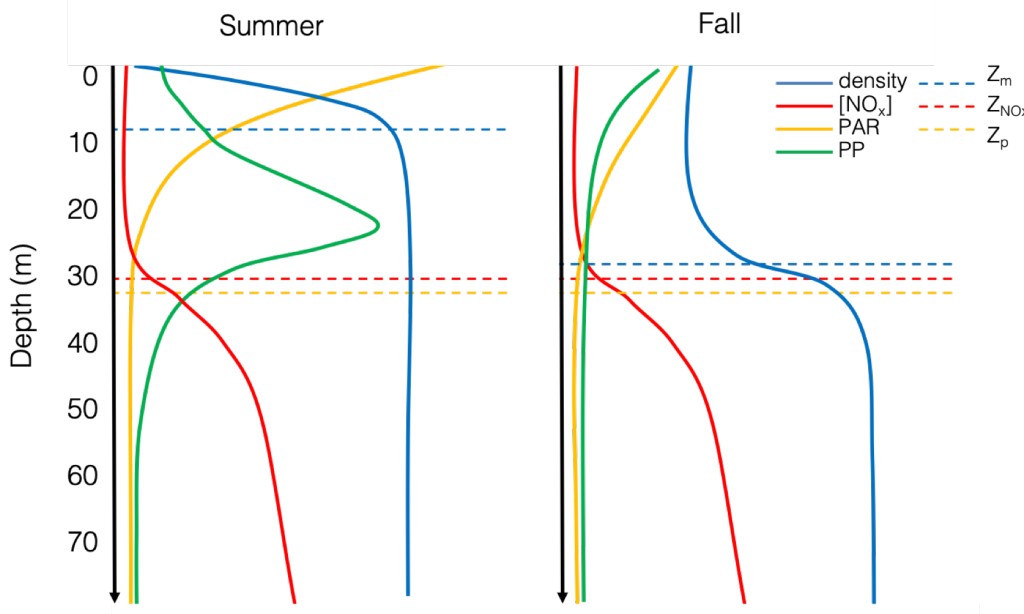