# Peer review of "Seasonal and spatial patterns of primary production in a highlatitude fjord affected by Greenland Ice Sheet run-off"

_Biogeosciences, 2019_

## Referee Comment (RC1) · Jose Iriarte (Referee) · 30 Jun 2019

The manuscript presents carbon rates of phytoplankton assemblages (total and size-fractionated) at several stations in a land-terminating arctic fjord duirng summer and fall months. This arctic fjord system is experiencing rapid environmental changes (climatic, hydrological), thus affecting oceanographic features such as halyne-driven stratification from ongoing freshening. Major results were: low primary production (PP, 14C) and autotrophic biomass (chl-a) were observed at inner stations associated to melting runnoff during summer months: during that period, an interplay between light and nutrient limitation (caused by particles and nitrogen, respectively) were raised as main factors for

the low PP estimates. This study provide an excellent base-line of carbon estimates dataset to understand the role of drivers (i.e., freshening process) at a global scale in fjord systems.

Specific comments - This study is focused in total and size-fractionated PP as well as chlorophyll-a. Major changes of variability could be at the taxa/species level instead: do the authors have phytoplankton abundance, richness or taxonomical analyses? It would be great to have some feeling of what kind of groups (at least) are dominating at the different study seasons and stations. The taxonomical analyses could fit very nicely to phytoplankton size classes.

- Regarding the above, could the authors explain what was the criteria for the phytoplankton size fractionation protocol; according to Sieburth et al, phytoplankton community can be splitted in three different size classes which match taxonomical groups: for example: picoplankton (mainly cyanobacteria: <2.0 um); nanoplankton (mainly flagellates: 2 – 20 um), microphytoplankton (a mix of diatoms and dinoflagellates species > 20 um). Is there any previous evidence on the size classes dominating the systems such as components of the microbial loop?

- In line 178 "in three different size fractions: >10 $\mu$m, GF/F (nominal pore size: 0.7 $\mu$m) to 10 $\mu$m and < 0.7 $\mu$m" . . . I was confused with the definition of the last size fraction: What group was collect in the filter? If water is passing through 0.7 um, what is the next filter to collect phytoplankton cells less tan 0.7 um? - In figure 4, in terms of chlorophyll-a in stations 3 and 4, diatoms or nanoflagellates dominated the system? whereas, nanoflagellates or picoplankton dominated at inner stations? The same for PP size fractionation. - Inorganic nutrients: Do authors have a knowledge of environment N:P:Si ratio for the fjord area during their sampling to search for spatial/temporal gradients on nutrients ratios other than actual concentrations? The manuscript present data on N:P ratio less than 16 (according to Redfield ratio) to infer PP "limitation"; N:Si (1:1) is another interesting ratio to explore in the near surface layer, especially for diatoms, a groups that needs silicic acid for the frustule. Again, species/functional

groups could respond more to ratios than concentrations; for example nanoflagellates respond better to N sources (ammonia), whereas diatoms could respond better to silicic acid concentrations

- Do the authors performed a nutrient limited experiment to infer "limitation"?. I would suggest to use "nutrient deficiency" instead.

- Since Young Sound is affected by sea-ice in the spring and run-off (river and or glacier) in summer, is there any information on the supply of inorganic nutrients from theses sources to the inner area of the fjord? Are sea-ice and run-off rich in any dissolved (micro-macro) nutrients?

- the first effect of increasing freshwater run-off is the vertical stratification of the water column; however, run-off could also explain density gradient circulation along the fjord, thus increasing its "ventilation" and probably generating internal waves traveling along/across the fjords that could be important to "break" the picnocline and bring inorganic nutrients to the surface "brackish" layer. Is there any possibility for this mechanism that could act fueling PP in this fjord in the near future?

- Role of wind: at the end of the period strong winds were evident; however there was no differences in the stratification index between seasons; according to literature, strong winds deepen the mixed layer and then lowering PP through the photic layer; it means that phytoplankton cells are spending more time below the photic layer. Any chance to use another SI that take mixed layer in consideration? I guess all CTD casts and PP incubations sampling were taking before and after stormy conditions.

- Figures and relationships: in figure 3, would be nice to have the PP sampling dates incorporates in the figure to see wind and run-off conditions. - in figure 4, I suggest a relationship with mean (±SD) integrated PP estimates to strengthen figure 6 conceptual model. - Any possibility to give some estimates of the assimilation index (integrated carbon uptake rates to integrated chl-a concentrations, mg C (mg chl-a)-1 day-1) which could provide a useful indicator of the potential physiological status of phytoplankton

(by size classes) along the freshwater influence gradient of the fjord.

---

## Referee Comment (RC2) · Anonymous Referee #2 · 10 Jul 2019

Summary: This manuscript presents a field based study on primary production in a high-Arctic Greenland fjord influenced by run-off from melting of glaciers. Phytoplankton carbon content and rates was measured on a temporal (summer and fall) and spatial (from fjord head to mouth) grid. The authors found that the overall production in the fjord is low but steady compared to similar fjords. Spatially the inner stations had a lower primary production and chl a concentration compared to the outer stations. These findings were attributed to melting run-off from the glacier, which reduces light (due to sediments) and nutrients. This study provides a very good baseline study for glacier-influenced fjords along the little studied northeast Greenland coast and adds to the growing number of work on primary production studies across the high-Arctic.

[Figure]

Such studies are of particular importance in a time of global change.

Specific comments: The effect of wind is often strong within fjords, both on average through the year and due to storm situations. You mention that a storm did appear during your study and usually strong winds and storms will affect the dynamics of the upper water column. Do you have any data that shows if the physical and primary productive dynamics changed in the water column after the storm?

In this study primarily carbon content, chl a etc. is measured and was shown to vary spatially and temporally. It is known that there is a succession of phytoplankton present through the year and that they have different production rates, are of different sizes etc. I therefore wonder if you have any data on the community of phytoplankton in the different samples/stations?

Technical corrections: Line 67: delete "that". L 97: remove the capital H in high-Arctic. 100 – this general paragraph: just out of curiosity could you add the depth at each of the stations? L 295: remove "a majority of" and replace with "primarily". L 361: remove "is that". Figure 1: Maker the numbers in Fig. 1a larger. The figure is generally small making the numbers difficult to spot. Moreover, is the paper is printed in black and white they become impossible to see.

---

## Author Comment (AC1) · 31 Jul 2019

Author response on "Seasonal and spatial patterns of primary production in a high-latitude fjord affected by Greenland Ice Sheet run-off" by Johnna M. Holding et al.

First, we would like to thank both referees for their time to provide insightful comments and for their contribution to the improvement of our manuscript.

Below is a detailed response to the comments from each of the two referees. Briefly, the main comments related to an interest in plankton community succession over the seasons studied. Furthermore, both referees were also interested in the dynamics in

the physical environment and effects on primary productivity during a stormy period at the end of the sampling season. We have addressed both of these queries in detail in the comments below. Briefly, we have included some more detail about the phytoplankton community succession in the text of the manuscript citing both published and unpublished data. We have decided not to include the unpublished data in this manuscript as we feel that the data set is too large and would dilute the main message of the present manuscript. Regarding the stormy period we have included more information regarding primary production rates and sampling during this period.

The amendments that we have made to the manuscript have not affected the main conclusions of the study, thus we hope that with these minor changes, the manuscript will now be acceptable for publication in Biogeosciences.

Sincerely, on behalf of all authors,

Johnna Holding
* * *
Referee 1- Jose Iriarte (R1):

R1: The manuscript presents carbon rates of phytoplankton assemblages (total and size- fractionated) at several stations in a land-terminating arctic fjord during summer and fall months. This arctic fjord system is experiencing rapid environmental changes (climatic, hydrological), thus affecting oceanographic features such as haline-driven stratification from ongoing freshening. Major results were: low primary production (PP, 14C) and autotrophic biomass (chl-a) were observed at inner stations associated to melting runoff during summer months: during that period, an interplay between light and nutrient limitation (caused by particles and nitrogen, respectively) were raised as main factors for the low PP estimates. This study provides an excellent base-line of carbon estimates dataset to understand the role of drivers (i.e., freshening process) at a global scale in fjord systems.

R1: Specific comments - This study is focused in total and size-fractionated PP as well as chlorophyll-a. Major changes of variability could be at the taxa/species level instead: do the authors have phytoplankton abundance, richness or taxonomical analyses? It would be great to have some feeling of what kind of groups (at least) are dominating at the different study seasons and stations. The taxonomical analyses could fit very nicely to phytoplankton size classes.

AC: We agree that there is a variability in phytoplankton community composition both seasonally and spatially in the fjord system. We have abundance data of picophytoplankton and nanophytoplankton from flow cytometry and the relative abundance of the most abundant phytoplankton/cyanobacteria species based on plastidial 16S rRNA gene sequence data. We however do not have the abundance of micro-sized phytoplankton and dinoflagellates as the counts from lugol fixed samples came back with poor quality and therefore counts of microphytoplankton are not reliable. Briefly, the results show that the innermost station was dominated by picophytoplankton in terms of biomass; these proved to be highly diverse and also include members of diatoms as well as freshwater cyanobacteria. This is unique, as in many other Arctic studies (Lovejoy et al., 2007; Terrado et al., 2011) the pico-fraction is dominated by one single Micromonas strain. While we do think it is interesting to include this information, we are planning to include the phytoplankton counts and diversity data in a separate paper as we think the data deserve the proper attention that only a separate paper could provide. We can however include a short summary of the data, as we agree it is interesting when discussing primary production. Data on cyanobacteria has been previously published in Paulsen et al. (2017) and data on phytoplankton with a focus on diatoms was addressed by Krawczyk et al., (2015).

We have now included the following paragraph in the discussion (now lines 394-404):

"Low light adaption is also evidenced by the smaller cells that dominated in the inner fjord throughout the season (Fig. 5). Smaller cells have no cell walls, and more efficient pigment packaging that give them a higher affinity to light (Raven, 1998; Taguchi,

1976). Plastidial 16S rRNA gene sequence data and flow cytometry cell counts confirm that picophytoplankton dominate biomass in much of the inner fjord (unpublished data) with freshwater cyanobacteria comprising a large fraction of the phytoplankton community there (Paulsen et al., 2017). Surprisingly the picophytoplankton community was not dominated by the green algae Micromonas sp. as is often found in other Arctic regions (Lovejoy et al., 2007; Terrado et al., 2011), rather, the data suggests the pico-group was comprised of a mixture of freshwater cyanobacteria, diatoms, and different green algae, as in Sørensen et al. (2012). Larger cells were more present in the summer in the outer fjord (Fig. 5) and are dominated by large diatoms and also dinoflagellates (Krawczyk et al., 2015), but community diversity was higher in the inner fjord (unpublished data). A higher diversity of smaller cell sizes contributes a competitive edge when it comes to light utilization efficiency (Schwaderer et al., 2011)."

R1: Regarding the above, could the authors explain what was the criteria for the phytoplankton size fractionation protocol; according to Sieburth et al, phytoplankton community can be splitted in three different size classes which match taxonomical groups: for example: picoplankton (mainly cyanobacteria: <2.0 um); nanoplankton (mainly flagellates: 2 – 20 um), microphytoplankton (a mix of diatoms and dinoflagellates species > 20 um). Is there any previous evidence on the size classes dominating the systems such as components of the microbial loop?

AC: We agree it is standard to consider picophytoplankton (<2um), nanophyto (2-20um) and microphytoplankton (>20um) as functional groups of the microbial foodweb. However, primary production (PP) measurements were regrettably not estimated for the pico-fraction in this study. As explained in the previous comment the data regarding the more biological aspects of phytoplankton and their adaption in the fjord system is planned to be published in a separate paper.

R1: In line 178 "in three different size fractions: >10 $\mu$m, GF/F (nominal pore size: 0.7 $\mu$m) to 10 $\mu$m and < 0.7 $\mu$m" . . . I was confused with the definition of the last size fraction: What group was collect in the filter? If water is passing through 0.7 um, what

is the next filter to collect phytoplankton cells less tan 0.7 um?

AC: We thank the reviewer for pointing out something which is a bit unclear in the manuscript. The <0.7 $\mu$m fraction is actually representative of the "dissolved" fraction of primary production. To measure this fraction, a 10 mL sample was taken out of each bottle after incubation to measure the TOC (total organic carbon) production. The 10 mL sample was placed in a scintillation vial where it was acidified, and then scintillation cocktail was added prior to measuring on the counter. This method is a common way to measure TOC production, which is then subtracted from particulate production (in our case, that which was measured on a GFF filter) to estimate the dissolved fraction of production during the incubations. We have now refered to this fraction as "dissolved fraction" throughout the manuscript and amended the text in the methods (now lines 179-190) as follows to make this more clear:

"Furthermore, three Nunc bottles incubated at low, medium and high light were spiked with additional isotope, in order to measure production in three different size fractions: >10 $\mu$m, GF/F (nominal pore size: 0.7 $\mu$m) to 10 $\mu$m, and < 0.7 $\mu$m, hence forth referred to as the "dissolved fraction"). After the incubation, the total organic carbon production (TOC) was measured from a 10 mL sample taken from these three Nunc bottles and added to glass vials in which 500 $\mu$L 1 N HCl was added and the vials were gently bubbled 5 times over 48 hours before addition of 10 mL scintillation cocktail. The remaining ca. 52 mL in the three spiked Nunc bottles where filtered through 10 $\mu$m pore filters and the filtrate was collected. This filtrate, and the content of the other nine Nunc bottles, were filtered through GF/F filters. All filters were placed in plastic vials and acidified with 200 $\mu$L 1 N HCl. Then the vials were allowed to stand for 24 hours before they were closed and stored in a freezer. Within 1-2 months all vials were counted in a Perkin Elmer TriCarb 2910 TR scintillation counter. The 14C-uptake was calculated from the effect volume and the added amount of 14C, and carbon fixation was then calculated from the DIC-concentration. The dissolved fraction was calculated by subtracting the uptake on GFF filters from the TOC samples. PP fractions were

[Figure]

integrated over 100m in the same way as the chl a fractions."

We do not have a <0.7$\mu$m fraction for chl a as it would be 0 (largely all phytoplankton are > 0.7$\mu$m in Young Sound), we can confirm this from flow cytometer counts (unpublished data) as in our samples there are no Synechococcus or Prochlorococcus (i.e. the only phytoplankton that potentially can pass a 0.7$\mu$m filter).

R1: In figure 4, in terms of chlorophyll-a in stations 3 and 4, diatoms or nanoflagellates dominated the system? whereas, nanoflagellates or picoplankton dominated at inner stations? The same for PP size fractionation.

AC: Yes, indeed as the chlorophyll a fractions suggest in figure 5 the outer fjord was dominated by larger phytoplankton, while the inner fjord was mainly dominated by pi-cophytoplankton. This was stated in the results (now lines 302-304) and now also reiterated in the discussion (lines 394-404; see comments from above).

R1: Inorganic nutrients: Do authors have a knowledge of environment N:P:Si ratio for the fjord area during their sampling to search for spatial/temporal gradients on nutrients ratios other than actual concentrations? The manuscript present data on N:P ratio less than 16 (according to Redfield ratio) to infer PP "limitation"; N:Si (1:1) is another inter-esting ratio to explore in the near surface layer, especially for diatoms, a groups that needs silicic acid for the frustule. Again, species/functional groups could respond more to ratios than concentrations; for example nanoflagellates respond better to N sources (ammonia), whereas diatoms could respond better to silicic acid concentrations.

AC: We agree with reviewers, we could include more detail about nutrient ratios in the text of the manuscript. As the reviewer suggests, we have explored spatial and temporal gradients in nutrient ratios using a similar approach as Figure S1, now by separating out stations and seasons. The spatial analysis showed no deviations from the overall trends in Figure S1 when separating out stations both with N vs. P and N vs. Si (see Figure A below). The seasonal analysis showed only that N vs. Si in the fall is much less variable in surface water than in the summer (see Figure B below). This

is likely due to the high concentrations of Si that are present in the run-off water (See Paulsen et al., 2017). We have now included information about the overall N:Si ratio in the results section of the manuscript (now lines 260-265). The changes to the text are copied below.

Figure A. Nutrient ratios at all stations and depths sampled. Nitrate + nitrite ($\mu$M) versus phosphate ($\mu$M) concentrations (left), and nitrate + nitrite ($\mu$M) v. silicate ($\mu$M) concentrations (right). Open symbols indicate depths above the average nitracline of the data set (29m). Blue = station 1, red = station 2, green = station 3, purple = station 4. Black lines represent the 16N:1P and 15Si:16N Redfield ratios for phosphate and silicate respectively.

Figure B. Nutrient ratios at all stations and depths sampled. Nitrate + nitrite ($\mu$M) versus phosphate ($\mu$M) concentrations (left), and nitrate + nitrite ($\mu$M) v. silicate ($\mu$M) concentrations (right). Open symbols indicate depths above the average nitracline of the data set (29m). Orange = summer and Red = fall. Black lines represent the 16N:1P and 15Si:16N Redfield ratios for phosphate and silicate respectively.

We have amended the text in the manuscript as follows:

"Indeed, the average NOx to phosphate ratio (N:P) for the data set is 2.2 $\pm$ 2.6 (mean $\pm$ SD), much below the Redfield value of 16, as such, communities are deficient in nitrogen with phosphorous in surplus (Figure S1). Silicate increased from 4.35 $\pm$ 1.88 $\mu$M to 6.87 $\pm$ 0.75 $\mu$M at 100 m depth, though at the surface silicate concentrations are variable and range from 0.69 to 10.0 $\mu$M, due to high concentrations of silicate (range: $3-40$ $\mu$M) in the meltwater run-off (Paulsen et al. 2017). As such, the silicate to NOx ratio (N:Si) for the data set (mean $\pm$ SD: 0.28 $\pm$ 0.30) is variable but in generally much lower than the Redfield value of 1.07 (Figure S1)."

R1: Do the authors performed a nutrient limited experiment to infer "limitation"?. I would suggest to use "nutrient deficiency" instead.

AC: Indeed, we have not performed nutrient limitation experiments, we have changed the word "limitation" to "deficiency" in regards to nutrients throughout the manuscript.

R1: Since Young Sound is affected by sea-ice in the spring and run-off (river and or glacier) in summer, is there any information on the supply of inorganic nutrients from theses sources to the inner area of the fjord? Are sea-ice and run-off rich in any dissolved (micro-macro) nutrients?

AC: A detailed account of nutrient concentrations in run-off water in Young Sound during the same year are previously reported in Paulsen et al. (2017). They are:

Nitrate + nitrite (NOx): 0.06 – 1.7 $\mu$M Phosphate: 0.08 – 0.6 $\mu$M Silicate: 3 – 40 $\mu$M

We have now included that information in the text of the results (lines 262-268):

"Silicate increased from 4.35 $\pm$ 1.88 $\mu$M to 6.87 $\pm$ 0.75 $\mu$M at 100 m depth, though at the surface silicate concentrations are variable and range from 0.69 to 10.0 $\mu$M, due to high concentrations of silicate (range: 3 – 40 $\mu$M) in the meltwater run-off (Paulsen et al. 2017). As such, the silicate to NOx ratio (N:Si) for the data set (mean $\pm$ SD: 0.28 $\pm$ 0.30) is variable but in generally much lower than the Redfield value of 1.07 (Figure S1). While the rivers enrich the surface water with silicate, NOx (range: 0.06–1.7 $\mu$M) and phosphate (0.08–0.6 $\mu$M) concentrations in river run-off are generally in the range of surface water concentrations for these nutrients. For further details of nutrient profiles in the water column and run-off see Fig. S1 in Paulsen et al. (2017)."

Sea-ice it is generally not considered a major source of nutrients to the surface ocean, however, there are some measurements of NOx in the sea ice in Young Sound which report 1,400 $\mu$mol N m-2 (Rysgaard and Glud, 2004). Much of this, however, is lost in situ by either denitrification or primary production in the ice matrix, as evidenced by nitrogen profiles in the sea ice which are reduced to below surface water concentrations just before ice melt (Glud et al., 2007).

R1: the first effect of increasing freshwater run-off is the vertical stratification of the

water column; however, run-off could also explain density gradient circulation along the fjord, thus increasing its "ventilation" and probably generating internal waves traveling along/across the fjords that could be important to "break" the picnocline and bring inorganic nutrients to the surface "brackish" layer. Is there any possibility for this mechanism that could act fueling PP in this fjord in the near future?

AC: In short, yes, internal wave activity has the potential to mix the water column in Young Sound during the summer and fall and likely plays a role in exchange of surface and intermediate waters in the fjord at present (Boone et al. 2018), however, there has yet to be an analysis of the relevant data to determine if this behavior will increase or decrease in the future. We have now mentioned internal waves in the discussion as a possible driver of mixing (now lines: 449-456).

"Lack of mixing processes during summer also have limited capacity to replenish nitrate to the photic zone in Young Sound. First, the maximum tidal amplitude in Young Sound is only 0.8 m (Bendtsen et al., 2007), which is low compared to the tidal amplitude in the more productive Godthåbsfjord of up to 5 m (Blicher et al., 2013). Additionally, in Young Sound, river discharge throughout the summer creates a surface lens of freshwater and hence mixed layer depth shallower than 10m, but in the fall when river run-off ceases, the mixed layer depth deepens to approximately 30m where the nitracline also sits (Fig. 6). In that period, nutrients may be mixed up via estuarian mixing and internal waves generated at the sill (Boone et al., 2018; Cottier et al., 2010) bringing up some nutrients, which may account for the low but significant rates of primary production during that time period."

R1: Role of wind: at the end of the period strong winds were evident; however there was no differences in the stratification index between seasons; according to literature, strong winds deepen the mixed layer and then lowering PP through the photic layer; it means that phytoplankton cells are spending more time below the photic layer. Any chance to use another SI that take mixed layer in consideration? I guess all CTD casts and PP incubations sampling were taking before and after stormy conditions.

AC: We agree with the reviewer, there was a peak in wind activity in the very end of September/ beginning of October (see Figure 3b in the present manuscript). This occurred just before the very last sampling days on October 2-6th. As we use only a small boat for sampling, we could only go out before and after these wind conditions. The effects of the strong wind and deep mixing can be seen clearly in the last length transect of the fjord on October 4th (Figure 1). The contour plots of fluorescence and salinity show deep mixing down to at least 80m (the mixing depth in station 4 on the last sampling day; Table 1). Furthermore, station 4 also exhibits a much lower stratification index (0.68) on the last sampling day (Table 1). On the other hand, from figure 1, wind mixing is not evident inside the fjord, starting at station 3, though in figure S2 the distribution of chlorophyll in the upper water column shows a bit deeper distribution (to ~40m) suggestive of deeper mixing. We did not find a better estimate of stratification index that incorporates mixing depth, rather, typical calculations of SI consider temperature stratification instead of salinity stratification, which is why we chose a method specific to the Arctic which takes into account the density of upper and lower water masses (See (Tremblay et al., 2009). While it doesn't take the actual mixing depth into account it is assumed (as we can clearly see in the contour plots of Figure 1, that density at 80m is constant, which is why we can see such a change in mixing depth on the last sampling date in station 4. We have now included further discussion of the stratification index at station 4 on the last sampling date in the results and discussion in the following parts of the manuscript:

Lines 250-253: "In the fall, florescence was more homogeneous throughout the upper water column (Fig. 1e), and the DCM moved higher up in the water column in all stations (fall mean $\pm$ SD DCM at main sampling stations: $18 \pm 11$) except for the outer-most stations, which were subject to deep mixing (DCM: 69 m) from high winds that took place at the end of the September (Fig. 3b)."

Lines 273-277: "The mixed layer depth (Zm) increased over the season at all stations (Fig. 4a; Table 1) from a mean ($\pm$ SD) of $9.3 \pm 4.6$ m in the summer to $27.1 \pm 5.5$ m in

the fall excluding the last sampling date of Station 4 which had an unusually deep Zm (86m). Stratification index (SI) was not significantly different between seasons (mean ± SD SI in summer and fall respectively: 6.3 ± 3.5 and 3.4 ± 0.9 kg m-3), though it did decline to 0.68 in station 4 on the last sampling day on October 4threlated to deep wind mixing from the previous days' storm (Fig. 3b)."

Lines 456-463: "Though patterns of stratification are very different between seasons the water column remains stratified well into the fall. There was some increase in wind speed, as well as a large storm towards the end of the season (Fig. 3b). The stratification index in fall tended to be slightly lower than in the summer though not significantly different, except in station 4 on the last sampling day in October after the storm had past where the value was much lower (Table 1). SI values in this study are 2-3 times higher than SI values calculated in the same way in Labrador fjords also influenced by river run-off (Simo-Matchim et al., 2016). This suggests that in the absence of mechanical mixing, any increase in the stratification index due to freshening could reduce vertical mixing even further."

R1: Figures and relationships: in figure 3, would be nice to have the PP sampling dates incorporates in the figure to see wind and run-off conditions.

AC: We agree with the reviewer and have now included dashed lines in the figures to indicate the summer and fall sampling periods.

R1: in figure 4, I suggest a relationship with mean (±SD) integrated PP estimates to strengthen figure 6 conceptual model.

AC: We agree with the reviewer that a more simplistic view of the integrated PP estimates in the style of Figure 4 would be helpful in visualizing our conclusions. We present below Figure C,a, which shows integrated PP measurements over time for each station. We now included it in the manuscript as the new Figure 6, together with a similar figure representing the specific primary production over time which addresses the reviewer's comments below as well.

Figure C. Areal primary production (mgC m-2 day-1; a) and specific areal primary production (mgC mgChl a-1 day-1; b) for each station over the length of the sampling season.

R1: Any possibility to give some estimates of the assimilation index (integrated carbon uptake rates to integrated chl-a concentrations, mg C (mg chl-a)-1 day-1) which could provide a useful indicator of the potential physiological status of phytoplankton (by size classes) along the freshwater influence gradient of the fjord.

AC: We agree, and we have now included an areal PP/chl a figure (see comment above) as the new figure 6 in the manuscript. In general, patterns are similar to the ones we see in Figure C,a above, however, there are a few major differences that we can point out. Station 1 in the summer in the inner fjord has much higher rates of primary production per unit biomass than Station 3 in the summer during the under-ice bloom. And with the exception of the bloom in station 3, in general stations 1 and 2 in the inner fjord have higher rates of PP per unit chl a than the outer fjord stations. This is likely also an indicator of the adaption of the inner fjord communities to low light as they are more efficient per unit chlorophyll than the other fjord communities. Furthermore, data from flow cytometry—-which will be the topic of a separate paper and thus are not included here—-show that the fluorescence per unit biomass is also much higher in the inner fjord communities, so these smaller cell sized communities have functionally adapted to their low light environment. We have now included the following paragraph in the discussion (now lines 404-413):

"Finally, another indication of low light adaptation is seen in the specific primary production; that is the areal primary production per unit chlorophyll (mgC mg Chl a-1 day-1; Fig. 6b). In general, seasonal patterns are similar to the ones we see in areal primary production (Fig. 6a), however it is notable that in the summer in the inner fjord (Station 1), there are much higher rates of primary production per unit Chl a than in Station 3 where there highest rate of areal production took place under the ice on the first sampling day. And, with the exception of the high under-ice production in Station

3, in general Stations 1 and 2 in the inner fjord have higher rates of specific primary production than the outer fjord stations. Thus, these smaller cell sized communities have functionally adapted to their low light environment and are more efficient per unit chlorophyll than outer fjord communities."

Anonymous referee 2 (R2):

R2: Summary: This manuscript presents a field-based study on primary production in a high-Arctic Greenland fjord influenced by run-off from melting of glaciers. Phytoplankton carbon content and rates was measured on a temporal (summer and fall) and spatial (from fjord head to mouth) grid. The authors found that the overall production in the fjord is low but steady compared to similar fjords. Spatially the inner stations had a lower primary production and chl a concentration compared to the outer stations. These findings were attributed to melting run-off from the glacier, which reduces light (due to sediments) and nutrients. This study provides a very good baseline study for glacier-influenced fjords along the little studied northeast Greenland coast and adds to the growing number of work on primary production studies across the high-Arctic. Such studies are of particular importance in a time of global change.

R2: Specific comments: The effect of wind is often strong within fjords, both on average through the year and due to storm situations. You mention that a storm did appear during your study and usually strong winds and storms will affect the dynamics of the upper water column. Do you have any data that shows if the physical and primary productive dynamics changed in the water column after the storm?

AC: We agree with the reviewer that there needs to be more information about the storm that occurred in the fall. Reviewer 1 had similar comments and we address them in a previous comment from reviewer 1 above. In summary, we highlight several areas in the manuscript were we now discuss the physical and biological consequences of the storm that took place before the last sampling dates.

R2: In this study primarily carbon content, chl a etc. is measured and was shown to

vary spatially and temporally. It is known that there is a succession of phytoplankton present through the year and that they have different production rates, are of different sizes etc. I therefore wonder if you have any data on the community of phytoplankton in the different samples/stations?

AC: We agree that it is interesting to know about the composition of the phytoplankton community spatially as well as seasonally. As this comment is similar to review 1's comments on the same matter, we refer the reviewer to responses and actions taken from above.

R2: Technical corrections:

Line 67: delete "that" AC: Done

L 97: remove the capital H in high-Arctic AC: Done

100 – this general paragraph: just out of curiosity could you add the depth at each of the stations? AC: Yes, we have now included the depth at each station. See lines 104-115.

L 295: remove "a majority of" and replace with "primarily" AC: Done

L 361: remove "is that" AC: Done

Figure 1: Maker the numbers in Fig. 1a larger. The figure is generally small making the numbers difficult to spot. Moreover, is the paper is printed in black and white they become impossible to see. AC: Done

References

[revised manuscript text omitted]

Please also note the supplement to this comment:
https://www.biogeosciences-discuss.net/bg-2019-203/bg-2019-203-AC1-supplement.pdf

———————————————————

Figure A.

[Figure]

**Fig. 1.**

Figure B.

[Figure]

**Fig. 2.**

Figure C.

[Figure]

**Fig. 3.**